# The Impact of Thallium Exposure in Public Health and Molecular Toxicology: A Comprehensive Review

**DOI:** 10.3390/ijms25094750

**Published:** 2024-04-26

**Authors:** Yung Chang, Chih-Kang Chiang

**Affiliations:** 1Graduate Institute of Toxicology, College of Medicine, National Taiwan University, Taipei 100233, Taiwan; d06447001@ntu.edu.tw; 2Department of Integrated Diagnostics & Therapeutics, National Taiwan University Hospital, Taipei 100225, Taiwan

**Keywords:** thallium, emerging contaminants, public health, food intake, toxicity

## Abstract

This review offers a synthesis of the current understanding of the impact of low-dose thallium (Tl) on public health, specifically emphasizing its diverse effects on various populations and organs. The article integrates insights into the cytotoxic effects, genotoxic potential, and molecular mechanisms of thallium in mammalian cells. Thallium, a non-essential heavy metal present in up to 89 different minerals, has garnered attention due to its adverse effects on human health. As technology and metallurgical industries advance, various forms of thallium, including dust, vapor, and wastewater, can contaminate the environment, extending to the surrounding air, water sources, and soil. Moreover, the metal has been identified in beverages, tobacco, and vegetables, highlighting its pervasive presence in a wide array of food sources. Epidemiological findings underscore associations between thallium exposure and critical health aspects such as kidney function, pregnancy outcomes, smoking-related implications, and potential links to autism spectrum disorder. Thallium primarily exerts cellular toxicity on various tissues through mitochondria-mediated oxidative stress and endoplasmic reticulum stress. This synthesis aims to shed light on the intricate web of thallium exposure and its potential implications for public health, emphasizing the need for vigilant consideration of its risks.

## 1. Introduction

Thallium (Tl) is a blue-white heavy metal of the rare earth group (atomic number 81, atomic mass 204.38), also known as a structural biochemistry/post-transition metal. It is soft and ductile, with an appearance similar to tin, and it exists in two oxidation states (I and III), with (Tl^+^) being more stable than (Tl^3+^). Thallium is primarily found in sulfide minerals associated with low-temperature hydrothermal mineralization, with as many as 89 known thallium minerals recorded [1]. Thallium has a wide range of applications, including in semiconductors, scintillation camera imaging, optical fibers glass, high-temperature superconducting materials, and coatings [2,3]. In recent years, numerous countries have experienced thallium contamination incidents, primarily attributed to the extraction and metallurgical activities of sulfide ores [4,5]. The industrial processes have led to the dispersion of dust, vapors, or liquid contaminants into the surrounding air, water sources, and soil, subsequently infiltrating the biosphere [6].

The data provided in this review were obtained through searches conducted using the commonly used scientific research engine Web of Science, accessed at https://clarivate.com/webofsciencegroup/solutions/web-of-science/ on 4 January 2024. During the retrieval process, the keyword “thallium” was used under the condition of “all fields”. The search was filtered within the Web of Science categories “Public Environmental Occupational Health”, “Environmental Sciences”, and “Toxicology”, respectively, and articles were carefully reviewed in chronological order. Preference was given to the latest papers, and articles were selected based on the presence of keywords in the title or abstract. Furthermore, relevant references cited in the articles were explored to further extend the search. In total, approximately 1500 articles were reviewed, and 78 articles were recruited.

## 2. Thallium Exposure in Daily Life

### 2.1. Thallium in Beverages

Recent research has revealed thallium contamination in drinking water in many countries and regions [7,8]. Tea leaves grown in different regions have also been found to be affected by thallium contamination [9]. Furthermore, the presence of thallium has been detected in tea prepared under various conditions [10]. In addition, the presence of thallium has also been detected in coffee beans [11]. Table 1 provides a compilation of the latest studies on this topic.

### 2.2. Thallium in Tobaccos

There are approximately 1.1 billion smokers globally, making the thallium content in tobacco an important public health issue [12]. Furthermore, non-smokers may be exposed to secondhand and thirdhand smoke, highlighting the importance of the investigation [13]. In recent years, the presence of thallium in tobacco has been discovered, and there is a correlation between its presence and the cultivation location [14,15]. Interestingly, counterfeit cigarettes were found to contain higher levels of thallium compared to authentic brands [16]. Previous research has indicated a significant difference in thallium content when comparing the residences of smokers and non-smokers [13]. Table 2 provides a compilation of the latest studies on this topic.

### 2.3. Thallium in Vegetables

After thallium contaminates the soil, it can be absorbed by plants [17]. Previous studies have indicated that thallium is present in vegetables from various regions, with the type of plant being a significant factor influencing absorption. Studies indicate that, based on estimated transfer factors and bioconcentration factors, watercress, snow peas, chard, and pak choi exhibit a high enrichment capability for thallium [18,19,20]. In summary, thallium exposure to the human body may occur through various pathways. Table 3 compiles data that exceed the maximum permissible level (MPL) [21].

## 3. Epidemiological Investigation of Thallium

### 3.1. The Effects of Thallium on Human Kidney Health

Chronic kidney disease (CKD) has become a serious public health problem, and exposure to heavy metals is a risk factor for CKD [22]. Previous studies have shown a significant relationship between kidney function and Tl (Table 4). A study was conducted on 5037 elderly individuals aged over 65 in Yinchuan City, Ningxia, China. Using an eGFR < 80 mL/min/1.73 m^2^ as the standard, a total of 1631 individuals exhibited abnormal eGFR values, and there was a negative correlation between Tl content and abnormal eGFR [23]. Similar results were also observed in other studies. A study of 934 patients with essential hypertension in Wuhan, China, revealed a positive correlation between Tl content in the urine of male patients and eGFR [24]. Interestingly, a study involving 592 elderly individuals aged over 60 with diabetes in Fuyang City, Anhui Province, China, indicated a negative correlation between Tl concentration in urine and CKD [25]. Furthermore, a survey of the urine of 512 adolescents aged 11–16 years in Mexico revealed a positive correlation between urine Tl concentration and eGFR [26]. Based on geographical proximity, a survey was conducted on 2069 residents living near the petrochemical plant in Yunlin, Taiwan. The results showed that, in comparison to the low-exposure group residing farther from the petrochemical plant, the high-exposure group had higher urinary thallium concentrations, significantly reduced eGFR, and a higher risk of developing CKD [27], which means that Tl enters the human body due to industrial activities. To summarize, there is a notable correlation between Tl exposure and renal function. A total of 27,733 participants aged over 20 from the National Health and Nutrition Examination Survey conducted between 2003 and 2012 were analyzed. The results showed that decreased renal function was related to decreased excretion of Tl in urine [28]. Tl in urine may become a biomarker of kidney function.

### 3.2. The Effects of Thallium on the Health of Children and Pregnant Women

Heavy metals pose a significant risk to fetuses [29], with Tl being able to enter the fetus through the maternal placenta [30,31]. Previous studies have shown that pregnant women and their children will be affected by Tl [Table 5]. Exposure of pregnant women to Tl is associated with an increased risk of preterm birth [32]. A study conducted in Anhui Province, China, found that umbilical cord serum Tl levels were associated with reduced height and weight in young girls [33]. The current study has unveiled a significant negative association between maternal Tl exposure during early pregnancy and cord blood leukocyte mtDNAcn [34]. Furthermore, maternal exposure to Tl during delivery is associated with shortened telomere length in newborns [35]. Tl not only affects the fetus but also alters the metabolic system of pregnant women. Urinary Tl shows a positive correlation with acetate, scyllo-inositol, formate, and dimethylamine. Conversely, it exhibits a negative correlation with trans-aconitate and N-acetyl neuraminic acid. Positive associations with pregnanolone-3-glucuronide were particularly robust in the first trimester, while negative associations were observed with estrogen metabolites in the third trimester [36]. It is worth noting that serum Tl concentration is significantly associated with the risk of gestational diabetes mellitus [37]. In summary, Tl presents potential risks for both children and pregnant women.

### 3.3. The Effects of Thallium on the Health of Smokers

Previous research has revealed the presence of thallium in cigarette components [13,14], further unveiling a clear correlation between thallium content and the specific brands and geographical origins of these tobacco products [15,16]. These findings emphasize the importance of investigating the impact of smoking populations and thallium content on public health. The study has reported a statistically significant elevation in the average urinary thallium levels among smokers compared to the control group [38,39]. Moreover, smokers exhibited higher serum thallium concentrations compared to non-smokers [40]. Notably, a significant association was observed between the duration of illicit opioid substance use and both urinary and hair thallium concentrations [38]. Higher urinary Tl levels are associated with poorer lung function [41]. Furthermore, another study has demonstrated an association between urinary Tl and the exacerbation of lung function decline, especially among smokers [42]. Research on secondhand and thirdhand smoke exposure has found a correlation between elevated levels of Tl and the residences of smokers [16]. To summarize, there exists a significant correlation between Tl exposure and lung function, with smokers being exposed to higher levels of Tl through cigarette smoking, potentially resulting in more severe lung damage. Table 6 provides an updated compilation of papers published on this subject.

### 3.4. Thallium Causes Psychological, Metabolic, and Other Effects

Research has indicated that environmental toxicants, especially heavy metals, are associated with autism spectrum disorders (ASD) [43,44,45]. Similarly, children diagnosed with ASD exhibited higher Tl levels on average [46]. Another study found similar results, indicating a positive correlation between urinary Tl levels in adults and ASD [47]. Moreover, brain research revealed that serum Tl levels are associated with subjective memory impairment and Alzheimer’s disease [48]. Regarding metabolism, urinary Tl levels are associated with a decrease in thyroid hormone (T4) levels [49]. Urinary Tl concentration is positively correlated with an increase in serum bilirubin levels, a marker of hepatic function [41]. In terms of body composition, urinary Tl is positively correlated with body mass index (BMI) and waist circumference [50]. It is worth noting that urinary thallium levels are significantly correlated with levels of the urinary DNA damage marker, 8-hydroxy-2-deoxyguanosine (8-OHdG) [51]. The effects of thallium exposure on the body are comprehensive and multifaceted, potentially posing risks to public health. Table 7 presents a summary of the psychological, metabolic, and other effects of Tl.

## 4. Molecular Toxicity and Adverse Reactions Caused by Thallium

The toxicity of thallium (Tl) is related to two of its chemical properties. Firstly, the toxic mechanism of Tl^+^ is associated with potassium (K^+^) ions [52]. The ionic radius of Tl^+^ is similar to that of K^+^ [53]. In the reduced state, the ionic radius of Tl^+^ is 1.76 Å, while that of K^+^ is 1.60 Å [54]. Tl^+^ will compete with K^+^ on potassium channels, interfering with potassium-dependent biological functions [2]. In aquatic toxicology research, the toxicity of Tl (I) depends on the concentration of potassium ions (K^+^) in freshwater environments [55]. The second aspect of Tl toxicity arises from its reaction with sulfhydryl (-SH) groups in proteins and mitochondrial membranes [2]. Thallium (Tl) exhibits a high affinity for sulfur (S) ligands, forming complexes with protein thiol groups [56]. In rats, Tl decreases the activity of Cu–Zn superoxide dismutase (SOD), and this characteristic is associated with it [57]. Furthermore, the toxicity of thallium (Tl) is similar to many heavy metals in that it inhibits heavy metal toxicity by binding to the thiol (-SH) groups on glutathione (GSH) [58,59].

### 4.1. Mitochondria-Mediated Oxidative Stress

The toxicity induced by thallium is comprehensive and wide-ranging, with the most common effect being the induction of oxidative stress. In the study of hepatotoxicity, the mitochondria isolated from rat Hepatocyte were cultured with Tl (I). The results indicate that Tl (I) induces a significant increase in the oxidative stress parameters of mitochondria, a decrease in the ATP/ADP ratio, mitochondrial membrane potential (MMP) collapse, and the release of cytochrome c [60]. Additionally, the study found that both Tl (I) and Tl (III) induce the formation of reactive oxygen species (ROS), lipid peroxidation, and mitochondrial membrane potential collapse in isolated rat liver cells [61]. Further mechanistic studies revealed that both Tl (I) and Tl (III) induce the cascade activation of caspase enzymes and the appearance of apoptotic phenotypes. However, treatment with mitochondrial permeability transition (MPT) pore closure agents (cyclosporin A and carnitine), ATP generators (L-glutamine, fructose, and xylitol), and antioxidants (α-tocopherol or deferoxamine) inhibited caspase-3 activation and apoptosis [62]. Previous reports have indicated that thallium (Tl) can pass through the blood–brain barrier, causing neurotoxicity. A series of studies were conducted using rat pheochromocytoma (PC12) cells as a model due to their similarities with sympathetic neurons. The results show that both Tl (I) and Tl (III) lead to concentration- and time-dependent decreases in cell viability, reduced glutathione levels, and increased levels of oxidants in the cytoplasm. Additionally, this results in a significant increase in mitochondrial H_2_O_2_ steady-state levels and a decrease in membrane potential [63]. Furthermore, it was discovered that the extent of cell apoptosis induced by Tl (I) and Tl (III) varies in the presence of epidermal growth factor (EGF). The results suggest that EGF partially mitigates toxicity by inhibiting the sustained activation of the MAPK signaling cascade, and indicate that p38 plays a crucial role as a mediator in Tl (III)-induced apoptosis in PC12 cells [64]. Using the HN9.10e cell line, which is a fusion of mouse hippocampal neuroblasts and neuroblastoma cells, it was observed that Tl (I) caused neurite shortening, loss of matrix adhesion, and an increase in cytoplasmic calcium. Additionally, dose-dependent increases in mitochondrial reactive oxygen species (mtROS) levels and decreases in transmembrane mitochondrial potential (ΔΨm) were observed [65]. In the exploration of neurotoxic mechanisms, exposure of E17-E18 Wistar rat primary hippocampal neurons to Tl (I) resulted in a significant decrease in cell viability, elevated levels of reactive oxygen species (ROS), and a notable increase in apoptosis rates. Electron microscopy revealed mitochondrial swelling and vacuolar degeneration. Additionally, the Nrf2-Keap1 signaling pathway demonstrated a protective role against Tl (I)-induced cytotoxicity, accompanied by a significant decrease in the level of the mitochondrial fusion protein Mitofusin 2 (Mfn2). Notably, the activation of the Nrf2-Keap1 signaling pathway through t-BHQ (Nrf2 and II phase detoxifying agent) maintained the functionality of Mfn2, contributing to the reduction of damage caused by Tl (I) [66]. To conclude, thallium induces mitochondrial-mediated oxidative stress, leading to damage, and subsequently triggers apoptosis. Table 8 provides an updated compilation of papers published on this subject.

### 4.2. Thallium Induces Cellular Toxicity by Eliciting Endoplasmic Reticulum Stress

As widely recognized, the Nrf2-Keap1 signaling pathway plays a crucial role in the unfolded protein response [67], and its activation provides protection against the toxicity induced by Tl (I) [66]. Tl (I) and Tl (III) induce increased expression of ATF-6 and IRE-1 in Madin–Darby Canine Kidney cells, leading to XBP-1 splicing and nuclear translocation, resulting in cellular endoplasmic reticulum stress [68]. In hepatopancreatic cells of the environmental pollution-sensitive indicator organism Gammarus pulex, exposure to Tl (I) resulted in increased numbers of lipid droplets, lysosomes, and autophagic vacuoles. Additionally, fragmentation and expansion of the rough endoplasmic reticulum (RER) were observed [69]. In summary, thallium induces endoplasmic reticulum (ER) stress. Table 9 provides an updated compilation of papers published on this subject.

### 4.3. Genotoxicity and other Adverse Effects of Thallium

In the study of genetic toxicity, exposure of human lymphocytes to Tl (I) and Tl (III) significantly increases structural chromosomal abnormalities with or without gaps, and raises the percentage of cells with abnormalities without gaps [70]. Exposure of human blood cells to Tl (I) increases the length of the comet assay [71]. Furthermore, Tl (I) causes a decrease in both the viability and motility of mature mouse sperm, along with an increase in fragmented DNA [72]. In mouse neuroblastoma cells (Neuro-2A), acetylcholinesterase activity is highly sensitive to inhibition by Tl [73]. In the study of immunotoxicity, Tl (I) suppresses the expression of genes involved in B cell development, enhances the production of thymic CD4^+^ T cells, and promotes the migration of initial CD4^+^ T cells and recent thymic emigrants (RTE) from the thymus to the spleen [74]. To summarize, thallium can induce DNA damage, leading to genotoxicity. Table 8 presents a summary of the molecular toxicity and adverse reactions of Tl. Table 10 provides an updated compilation of papers published on this subject.

### 4.4. The Toxicity and Adverse Reactions of Thallium in Mammalian Models

In rat experiments, Tl (I) leads to a reduction in glomerular filtration rate (GFR) and urine volume, an increase in proteinuria, and a decrease in plasma amino acid concentration, indicating kidney toxicity [75,76]. Tl (I) induces increased lipid peroxidation in the hypothalamus, cerebellum, and striatum, a decrease in GSH in the striatum, reduced Cu,Zn-SOD activity in the hypothalamus and striatum, and results in impaired motor function in the animals [57,77]. Exposure to Tl (I) during pregnancy in female CD-1 mice results in offspring with trunk curvature, tail abnormalities, rotation defects in the forelimbs and hindlimbs, and delayed skeletal ossification [78]. In summary, in addition to cellular-level experiments, thallium has also been found to induce varying degrees of toxicity in animal models, particularly evident in the kidneys and nervous system. Table 11 provides an updated compilation of papers published on this subject.

## 5. Conclusions

Thallium emissions from smelting plants and technology manufacturing processes enter rivers through wastewater and accumulate in soil, or they enter organisms through the air. Thallium exposure pathways include drinking water, beverages, cigarettes, and vegetables. Multiple epidemiological studies have found that thallium has a significant impact on renal function, with this phenomenon observed in elderly, young, or diseased populations. Children and pregnant women are highly sensitive populations. Research has found that thallium increases the risk of premature birth and impacts child development. Due to the presence of thallium in cigarettes, smokers have higher serum thallium concentrations, which are correlated with lung damage. Furthermore, it has been found that thallium levels in urine are associated with autism. Moreover, thallium is correlated with many metabolites including thyroid hormone (T4), bilirubin levels, and 8-hydroxy-2′-deoxyguanosine (8-OHdG), and it is even associated with parameters such as body mass index (BMI) and waist circumference. The toxicity induced by thallium is extensive. Although the mechanisms of thallium toxicity are not yet fully understood, the most common is mitochondria-mediated oxidative stress. Studies using various cell lines, including hepatocytes (rat), pheochromocytoma (rat), murine hippocampal neuroblasts, and primary hippocampal neurons, have found that thallium can cause mitochondrial damage, including increased oxidative stress, changes in mitochondrial membrane potential (ΔΨm), and decreased ATP/ADP ratio, and even induce cell apoptosis. Recent studies have found that thallium can induce cell toxicity by causing endoplasmic reticulum stress, including increased expression of ATF-6 and IRE-1, XBP-1 splicing, and nuclear translocation. It is worth noting that the Nrf2-Keap1 signaling pathway, which plays an important role in the unfolded protein response, can protect against Tl (I)-induced toxicity when activated. In vitro studies have shown that thallium can induce various adverse reactions, including genetic toxicity, decreased sperm motility, and abnormal migration of immune cells. Animal studies have found consistent results with the epidemiological findings: Tl (I) reduces glomerular filtration rate (GFR), urine volume, and plasma amino acid concentrations, while increasing proteinuria, leading to renal toxicity. Additionally, it increases lipid peroxidation in the brain and impairs motor function in animals, resulting in neurotoxicity. Exposure of mice to Tl (I) during pregnancy leads to developmental defects in offspring, indicating the presence of developmental toxicity (Figure 1). However, most countries have not established limits for Tl in agricultural products, animal feed, and human food, thereby increasing the potential public health risks.

## Figures and Tables

**Figure 1 ijms-25-04750-f001:**
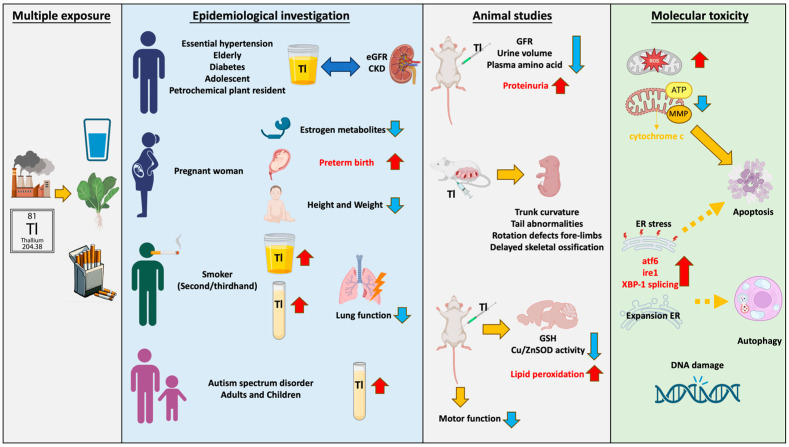
Thallium exhibits diverse contamination pathways and toxicity. Thallium pollution in the environment occurs through dust, steam, and wastewater. It enters organisms through sources such as drinking water, tobacco, and vegetables. Ingesting thallium in daily life can lead to damage in various organs and tissues. The red arrow pointing upwards indicates an increase; the blue arrow pointing downwards indicates a decrease; double arrows represent correlation; dashed arrows represent possible mechanisms.

**Table 1 ijms-25-04750-t001:** The content of thallium in beverages. The concentrations of thallium in drinking water, tea leaves, tea, and coffee beans in different regions.

Sample	Thallium Concentrations	Area	Ref.
Tap water (spring water)	10 (ppb)	Tuscany, Italy	[7]
Tap water (spring water)	27.8 (ppb)	Northwestern Tuscany, Italy	[8]
Tea leaves	4–409 (ppm)	Southwest, South, Jiangnan and Jiangbei tea regions of China	[9]
Soak tea	3.86 × 10^−4^ (ppm)	Anxi, China	[10]
Roasted coffee beans	2 ± 1 (ppb)	Brazil	[11]

**Table 2 ijms-25-04750-t002:** The thallium content within tobacco products. The concentration of thallium in tobacco across different regions and brands.

Thallium Concentrations	Area	Summary	Ref.
0.0089 ± 0.0012 (μg/g)	Poland		[13]
1.1 ± 0.1 to 2.4 ± 0.2 (ng/cigarette)	USA		[14]
0.70 ± 0.04 to 7.04 ± 0.23 (ng/cigarette)	USA	Mean smoke particulate thallium levels from counterfeit cigarettes were 1.4–4.9 times higher than authentic cigarettes.	[15]

**Table 3 ijms-25-04750-t003:** The thallium (Tl) content in vegetables exceeds the MPL (0.5 mg/kg). Concentrations of thallium in vegetables from different regions and varieties in China. MPL: maximum permissible level.

Vegetable	Tissue	Thallium Concentrations (mg/kg, Dry Weight)	Area	Ref.
Snow peas	In edible parts	1.47 ± 0.04	Guangdong Province, China	[18]
Napa cabbage	In edible parts	0.49 ± 0.01	Guangdong Province, China	[18]
Chard	In edible parts	1.65 ± 0.05	Guangdong Province, China	[18]
Scallions	In edible parts	0.77 ± 0.02	Guangdong Province, China	[18]
Taiwanese lettuce	In edible parts	0.92 ± 0.03	Guangdong Province, China	[18]
Pak choi	In edible parts	1.45 ± 0.04	Guangdong Province, China	[18]
Crown daisy	In edible parts	6.29 ± 0.26	Yunfu City, China	[19]
Baby bok choy	In edible parts	0.84 ± 0.03	Yunfu City, China	[19]
Romaine lettuce	In edible parts	0.49 ± 0.02	Yunfu City, China	[19]
Watercress	In edible parts	15.4 ± 0.51	Yunfu City, China	[19]
Gai lan	In edible parts	0.63 ± 0.02	Yunfu City, China	[19]
Oilseed rape	In edible parts	1.28 ± 0.05	Yunfu City, China	[19]
Sweet potato	In edible parts	2.78 ± 0.12	Yunfu City, China	[19]
Chinese mustard	In edible parts	0.56 ± 0.02	Yunfu City, China.	[19]
Bok choy	In edible parts	1.23 ± 0.05	Yunfu City, China	[19]
Green cabbage	In edible parts	0.85 ± 0.04	Yunfu City, China	[19]
Sonchus lingiaus	stems and leaves	0.72 ± 0.03	Pearl River, China	[20]
Allium fistulosum L	stems and leaves	0.92 ± 0.04	Pearl River, China	[20]

**Table 4 ijms-25-04750-t004:** The effects of Tl levels in urine on kidney functions in humans.

Subject Description	Subject Number	Specimen Type	Area	Summary of Effect	Ref.
Aged 65 years or older	37,523	Urine	Yinchuan, China	Tl was negatively correlated with abnormal eGFR.	[23]
Aged above 30 years old, essential hypertension	934	Urine	Wuhan, China	Tl was positively associated with eGFR in male participants.	[24]
Aged above 60 years, diabetic	592	Urine	Fuyang City, Anhui, China	Negative association between Tl concentrations and CKD	[25]
Adolescent 11–16 years old	512	Urine	Torreón, Mexico	Tl is positively correlated with eGFR.	[26]
Least 35 years old, living near a petrochemical complex	2069	Urine	Yunlin County, Taiwan	Decreased eGFR, higher odds of having CKD.	[27]

**Table 5 ijms-25-04750-t005:** Effects on human health in pregnant women and children associated with thallium (Tl) levels in urine, umbilical cord serum, and blood: overview of published studies.

Subject Description	Subject Number	Specimen Type	Area	Summary of Effect	Ref.
Mother–infant pairs	7173	Urine	Wuhan, China	Maternal exposure to Tl is related to increased risk of preterm birth.	[32]
Mother–infant pairs (Ages 0–2 year, fetus)	3080	Maternal serum/umbilical cord serum	Anhui, China	Umbilical cord serum Tl levels were associated with reduced height and weight in young girls.	[33]
Mother–newborn pairs	746	Uurine/umbilical cord blood	Wuhan, China	Maternal urinary Tl is associated with reduced mtDNAcn leukocytes in cord blood.	[34]
Mother–infant pairs	746	Urine/umbilical cord serum	Wuhan, China	Maternal exposure to thallium during delivery is associated with shortened telomere length in newborns.	[35]
Pregnant woman	750	Urine/blood/umbilical cord serum	Spain	Urinary thallium is positively correlated with scyllo-inositol, acetate, formate, dimethylamine. Urinary thallium is negatively correlated with N-acetyl neuraminic acid and trans-aconitate. Positive associations with pregnanolone-3-glucuronide were particularly strong in the first trimester and negative associations with estrogen metabolites in the third trimester.	[36]
Pregnant woman	3013	Blood	Anhui, China	Serum thallium concentration significantly associated with risk of gestational diabetes mellitus (advanced age).	[37]

**Table 6 ijms-25-04750-t006:** The effects on human health of smoker of Tl in urine/blood.

Subject Description	Smokers	Non- Smokers	Specimen Type	Area	Summary of Effect	Ref.
Age range of 23 to 77 years	56	53	Urine	Iran	The mean value (with SD) for urinary thallium in the smokers (10.16 ± 1.82 μg/L) was significantly higher than in the control group (2.39 ± 0.63 μg/L).	[39]
Age range of 19 to 74 years	100	100	Serum	USA	This study indicated that the level of thallium was higher in smokers than in non-smokers. Tl (0.54 [0.27–0.68] versus 0.34 [0.11–0.66] (μg/L), *p* = 0.04).	[40]
Age range of 21 to 81 years	50	50	Urine/blood/hair	Iran	There were significant correlations between duration of illicit opioid use and urine thallium concentrations (r = 0.394, *p* = 0.005) and hair thallium concentrations (r = 0.293, *p* = 0.039).	[38]
Age range of 10 to 15 years	27	46	Indoor/outdoor sampling of PM 2.5	Italy	Indoor smoking is associated with elevated Tl levels and is associated with increased respiratory symptoms in children.	[16]
Aged above 18 years	526	1837	Urine/serum	Central, east, northeast, north, northwest, south, and southwest China	Tl was negatively associated with increases in lung health indicators.	[41]
Workers from a coke oven plant	790	453	Urine	Wuhan, China	Urinary thallium is associated with increased lung function decline, especially in smokers.	[42]

**Table 7 ijms-25-04750-t007:** The psychological, metabolic, and other effects caused by thallium.

Subject Description	Subject Number	Specimen Type	Area	Summary of Effect	Ref.
Age 5–16 years old	44	Urine	Arizona, USA	Children with autism spectrum disorder (ASD), on average, had higher levels of thallium.	[46]
Age of 2.5 years to 60 years	67	Urine	Arizona, USA	Urinary thallium is positively correlated with autism spectrum disorder.	[47]
65.53 ± 6.37 years	118	Serum	Italy	Serum thallium correlates with subjective memory complaints and Alzheimer’s disease.	[48]
Aged above 20 years	1587	Urine	USA	Urinary thallium is associated with decreased Thyroxine (T4).	[49]
Aged above 18 years	2363	Urine/serum	Central, east, northeast, north, northwest, south, and southwest China	Tl > 0.40 μg/g was positively associated with increases in serum bilirubin.	[41]
6–60 years old	3816	Urine	USA	Thallium is positively associated with body mass index (BMI) and waist circumference (WC).	[50]
Aged 20 to 26 years	53	Urine	Guangzhou, China	The levels of Tl correlated significantly with the urinary 8-hydroxy-2-deoxyguanosine (8-OHdG) level.	[51]

**Table 8 ijms-25-04750-t008:** Thallium induces mitochondria-mediated oxidative stress and apoptosis.

Experimental Animals/Materials	Thallium Compounds	Concentrations	Summary of Effect	Ref.
Isolated rat liver (male Sprague Dawley)	Thallium (I) nitrate	25–200 μM	Increase in mitochondrial ROS formation, ATP consumption, GSH oxidation, mitochondrial outer membrane rupture, mitochondrial swelling, MMP collapse, and cytochrome c release. Effect dose: 25 μM.	[60]
Hepatocyte Male Sprague Dawley	Thallium (I) nitrate Thallium (III) nitrate	Tl (I) 200 μM Tl (III) 50 μM	Generation of reactive oxygen species (ROS), mitochondrial membrane potential collapse, activation of the caspase cascade, and the appearance of the cellular apoptosis phenotype. Effect dose: Tl (I) 200 and Tl (III) 50 μM.	[61]
Hepatocyte Male Sprague Dawley	Thallium (I) nitrate Thallium (III) nitrate	Tl (I) 200 μM Tl (III) 50 μM	Inducing ROS generation, lipid peroxidation, mitochondrial membrane potential collapse, activation of the caspase cascade, and lysosomal membrane leakage. Tl (I) 200 and Tl (III) 50 μM.	[62]
Rat adrenal pheochromocytoma (PC12 cells)	Thallium (I) nitrate Thallium (III) nitrate	10–250 μM	Tl(I) and Tl (III) significantly increase mitochondrial H_2_O_2_ steady-state levels. The glutathione content is significantly reduced in cells treated with Tl. There is a higher level of oxidants in the cytoplasm, which is positively correlated with the mitochondrial H_2_O_2_ content. Effect dose: 10 μM.	[63]
Rat pheochromocytoma (PC12)	Thallium(I) nitrate Thallium (III) nitrate	25–100 μM	Reduce mitochondrial membrane potential, enhance H_2_O_2_ generation, and activate mitochondria-dependent cell apoptosis. Tl (III) increases nitric oxide production, leading to an imbalance between anti-apoptotic and pro-apoptotic members of the Bcl-2 family. Effect dose: 25 μM.	[64]
Murine hippocampal neuroblasts (HN9.10e)	Thallium (I) chloride	1–100 μg/L	Neurite shortening, loss of substrate adhesion, and an increase in cytoplasmic calcium. Dose-dependent changes in mitochondrial ROS (mtROS) levels and transmembrane mitochondrial membrane potential (ΔΨm). Effect dose: 10 μg/L.	[65]
Primary hippocampal neuron E17-E18 Wistar rat embryos	Thallium (I) nitrate	100–200 μM	The Nrf2-Keap1 pathway can prevent thallium-induced oxidative stress and mitochondrial dysfunction. Effect dose: 100 μM.	[66]

**Table 9 ijms-25-04750-t009:** Thallium induces cellular toxicity by eliciting endoplasmic reticulum stress.

Experimental Animals/materials	Thallium Compounds	Concentrations	Summary of Effect	Ref.
Madin–Darby Canine Kidney cells	Thallium (I) nitrate Thallium (III) nitrate	Tl (I) 10,100 μM Tl (III) 10,100 μM	The expression of endoplasmic reticulum (ER) stress markers ATF-6 and IRE-1 increase by 100% and 150%, respectively, accompanied by XBP-1 splicing and nuclear translocation. Effect dose: 10 μM.	[68]
Gammarus pulex	Thallium (I) acetate	0.1–0.6 mg/L	Observations through transmission electron microscopy revealed the fragmentation and expansion of the rough endoplasmic reticulum (RER). Additionally, an increase in the quantity of lipid droplets, lysosomes, and autophagic vesicles was observed within liver cells. Effect dose: 0.2 mg/L.	[69]

**Table 10 ijms-25-04750-t010:** The genotoxicity and other adverse effects of thallium.

Experimental Animals/Materials	Thallium Compounds	Subject Number	Concentrations	Summary of Effect	Ref.
Human lymphocyte	Thallium (I) sulfate Thallium (III) chloride		0.5, 1, 5, 50, 100 μg/mL	Treatment with Tl (I) and Tl (III) significantly increases structural chromosomal abnormalities with or without gaps and raises the percentage of cells with abnormalities without gaps. Effect dose: 0.5 mg/L.	[70]
Human blood cells	Thallium (I) acetate		0.5, 1, 5, 10, 50, 100 µg/mL	Increased comet assay length. Mitotic and replication indexes exhibit a significant dose-dependent decrease. Effect dose: 0.5 mg/L.	[71]
Male mouse	Thallium (I)	Not disclosed	Adult male mice were administered Tl (15 or 25 mg/kg bw, orally, once a day for 5 days), and euthanized 24 h post-treatment.	Both sperm viability and motility decrease. DFI% (fragmented DNA) increases. Effect dose: 15 mg/kg bw.	[72]
Mouse neuroblastoma cells (Neuro-2A)	Thallium (I) acetate		0, 0.1, 1, 10, 100, 1000 mg/L	The activity of acetylcholinesterase is highly sensitive to Tl inhibition. Effect dose: 100 mg/L.	[73]
C57BL/6J male mice (specific-pathogen-free, SPF)	Thallium (I) nitrate	15	50 mg/L(drink). Euthanasia was performed after continuous exposure for one week.	Tl (I) increases apoptosis in bone marrow, suppresses the expression of genes involved in B cell development, enhances the production of thymic CD4^+^ T cells, and promotes the migration of initial CD4^+^ T cells and recent thymic emigrants (RTE) from the thymus to the spleen. Effect dose: 50 mg/L.	[74]

**Table 11 ijms-25-04750-t011:** The toxicity and adverse reactions of thallium in mammalian models.

Experimental Animals/Materials	Thallium Compounds	Subject Number	Concentrations	Summary of Effect	Ref.
Wistar rats (female)	Thallium (I) sulfate	6	5, 10, 15, 20 mg Tl_2_SO_4_/kg body weight, intraperitoneally. A single intraperitoneal injection is administered on either day 10 or day 55 after birth.	The glomerular filtration rate (GFR) and urine output decrease, while proteinuria increases. Effect dose: Tl, 2 mg/100 g b.wt.	[75]
Wistar rats (female)	Thallium (I) sulfate	6	5, 10, 15, 20 mg Tl_2_SO_4_/kg body weight, intraperitoneally. After a single dose of T1 administration, observations were made for up to 10 days.	The glomerular filtration rate (GFR) decreased and urine output reduced, while proteinuria increased. Effect dose: Tl, 20 mg/kg b.wt.	[76]
Male Wistar rats	Thallium (I) acetate	6	8 or 16 mg/kg i.p. At 1 day, 3 days, and 7 days post T1^+^ administration, rats were euthanized by decapitation, and their brains were rapidly removed.	After administration of Tl for 7 days, there is a dose-dependent accumulation observed in the brain region. Lipid peroxidation increases in the hypothalamus (Ht), cerebellum (Ce), and striatum (S), while the activity of Cu,Zn-superoxide dismutase (SOD) decreases in both Ht and S. Additionally, animals exhibit a general decline in motor function. Effect dose: 8 mg/kg.	[57]
Adult male bred-in-house Wistar rats	Thallium (I) acetate	6–10	Rats were administered T1 acetate solution intraperitoneally daily for 30 days at doses of 0.8 mg/kg or 1.6 mg/kg. On the third day after the treatment concluded, the animals were euthanized by decapitation.	Significant changes in lipid peroxidation occur in the rat’s brain. Effect dose: 0.8 mg/kg.	[77]
Sexually mature CD-1 mice	Thallium (I) acetate	10	A single intraperitoneal injection was administered to 10 pregnant female mice groups at doses of 4.6, 9.2, or 18.5 mg/kg body weight (bw) of TI(I) acetate.	Curvature of the trunk, tail abnormalities, rotation defects in the forelimbs and hindlimbs. Delayed skeletal ossification. Effect dose: 4.6 mg/kg.	[78]

## Data Availability

The data presented in this study are available from the corresponding author upon reasonable request.

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
