# Peer review of "The Impact of Thallium Exposure in Public Health and Molecular Toxicology: A Comprehensive Review"

_ijms, 2024, doi:10.3390/ijms25094750_

Round 1

Reviewer 1 Report

Comments and Suggestions for Authors

This is a well-organized and useful review that focuses on the long-term health effects of chronic exposure to low doses of thallium (Tl). By contrast, a recent detailed review on Tl toxicology (Fujihara and Nishimoto, Curr. Res. Toxicol. 2024, 6, 100157) focused primarily on the effects of acute Tl poisoning. Therefore, the two reviews are complementary. The review of Fujihara and Nishimoto should be referred to in the current work. Other suggested minor changes are as follows:

1.       The words ‘comprehensive’, ‘thorough’ and ‘meticulous’ in the Abstract should be avoided. These are the qualities that will be judged by the readers.

2.       The first paragraph of the Introduction contains numerous instances where a space should be inserted between a word and a bracket, such as ‘Thallium(Tl)’ and ‘group(atomic’ in line 27.

3.       Where possible, the values listed in tables, particularly in Table 1, should be in the same units, even if they are given in different units in the original works. A good option is to use the ppm and ppb units.

4.       The heading of Table 3 (lines 72-75) should be on the same page with the table body.

5.       The sentence starting with ‘The values for Tl’ (line 74) in the heading of Table 3 is out of place. This applies to Ref. 21 and should be either in a separate table or in the main text.

6.       Line 81: ‘Tl[Table4]’ should be ‘Tl (Table 4)’ (with a space and round brackets).

7.       Avoid the use of words ‘Papers published, which describe’ in the table headings. The table headings should be more streamlined, such as ‘The effects of Tl levels in urine on kidney functions in humans’ for Table 4.

8.       Line 128: ‘smokers’, not ‘smoker’.

9.       In Section 4, the authors should mention the two main chemical properties of Tl that determine its toxicity, namely its similarity with K and the high affinity to biological thiols (see the review of Fujihara and Nishimoto).

10.   Check the reference 21, it seems to be missing the authors’ names.    

Comments on the Quality of English Language

Some minor grammatic and stylistic issues are listed in the comments for authors.

Reviewer 2 Report

Comments and Suggestions for Authors

The English in this review article is generally good and does not significantly detract from the comprehensibility of the content. The authors demonstrate a strong command of technical terminology related to thallium exposure, toxicology, and epidemiology.

However, there are occasional grammatical errors, awkward phrasings, and areas where the language could be more concise and polished. For example:

  1. Some sentences are overly long and complex, which can hinder readability. Breaking them into shorter, clearer sentences would improve flow.
  2. There are a few subject-verb agreement errors.
  3. Some word choices are imprecise or overly casual for a scientific review, such as "in summary" being used repetitively as a transition.
  4. Occasional misuse of articles (a/an/the) and plural/singular nouns suggest the authors are not native English speakers.
  5. Some phrasing is awkward or unclear, such as "Epidemiological research indicates that an elevation in urinary Tl levels is inversely correlated with lung health indicators." This could be stated more directly as "Higher urinary Tl levels are associated with poorer lung function."
Comments on the Quality of English Language

The English in this review article is generally good and does not significantly detract from the comprehensibility of the content. The authors demonstrate a strong command of technical terminology related to thallium exposure, toxicology, and epidemiology.

However, there are occasional grammatical errors, awkward phrasings, and areas where the language could be more concise and polished. For example:

  1. Some sentences are overly long and complex, which can hinder readability. Breaking them into shorter, clearer sentences would improve flow.
  2. There are a few subject-verb agreement errors.
  3. Some word choices are imprecise or overly casual for a scientific review, such as "in summary" being used repetitively as a transition.
  4. Occasional misuse of articles (a/an/the) and plural/singular nouns suggest the authors are not native English speakers.
  5. Some phrasing is awkward or unclear, such as "Epidemiological research indicates that an elevation in urinary Tl levels is inversely correlated with lung health indicators." This could be stated more directly as "Higher urinary Tl levels are associated with poorer lung function."

Reviewer 3 Report

Comments and Suggestions for Authors

The review paper by Chang and Chiang aimed to offer a thorough synthesis of the current understanding of the impact of low-dose Tl on public health, specifically emphasizing its diverse effects on various populations and organs. The topic of the proposed paper is important. However, it would benefit from few minor corrections before being suitable for publication.

Firstly, briefly outline the methodology or approach used in the review, including any criteria for selecting studies or data sources (after the introduction section in the manuscript mention: keywords, searched databases, number of recruited papers, etc).

Secondly, throughout the paper, my primary concern, as will be demonstrated below, revolves around the tables and conclusion, which necessitate additional data and polishing.

Titles of the tables require enhancement. For instance, instead of " Papers published, which describe the effects on human health of pregnant women and children of Tl in urine/umbilical cord serum/blood," a more suitable title would be: "Effects on human health in pregnant women and children associated with thallium (Tl) levels in urine, umbilical cord serum, and blood: overview of published studies."

In all the tables, where applicable, add a column with measured parameters in the study.

For Table 6, please specify the number of subjects who are smokers and the number who are non-smokers.

Please enhance Tables 8-11 by including details regarding the duration and route of exposure for animal experiments. Additionally, please specify the doses at which the mentioned effects were observed.

This conclusion is too short. Although it briefly mentions the pathways of Tl exposure and the lack of established limits in various countries, it fails to adequately summarize the findings regarding the effects of Tl (observed in human, in vitro and in vivo studies). Additionally, what are the main mechanisms behind the observed effects? This should be summarised and added into the conclusion. For example, the authors have noted that epidemiological studies have identified thallium's impact on various population groups and organs, referencing Figure 1. However, it is crucial for the conclusion to explicitly specify which population groups and organs are affected. Furthermore, while Fig. 1 provides valuable insights, it lacks detailed information about the underlying mechanisms. Some of the clipart used in the figure may not sufficiently convey the intended meaning or be easily interpretable without accompanying textual explanations.
